# Inflammation—A Possible Link between Myocarditis and Arrhythmogenic Cardiomyopathy

**DOI:** 10.3390/diagnostics14030248

**Published:** 2024-01-24

**Authors:** Ioan Radu Lala, Adina Pop-Moldovan

**Affiliations:** 1Departement of Cardiology, Arad Emergency Clinical County Hospital, 310037 Arad, Romania; pop-moldovan.adina@uvvg.ro; 2Department of Cardiology, Arad Western University Vasile Goldis, 310414 Arad, Romania

**Keywords:** inflammation, myocarditis, arrhythmogenic cardiomyopathy, cardiovascular magnetic resonance

## Abstract

Arrhythmogenic cardiomyopathy is a primary genetic disease caused by mutations in the desmosome genes. Ever since the introduction of new imaging techniques, like cardiovascular magnetic resonance, the diagnosis of arrhythmogenic cardiomyopathy has become more challenging as left ventricular or biventricular variants may have resemblance to other cardiomyopathies or myocarditis. Not only this but they may also share an acute phase, which might cause even more confusion and misdiagnoses and influence the prognosis and outcome. In this case report, we present a 31-year-old patient with multiple clinical pictures: his symptoms were acute chest pain, new onset of heart failure and arrhythmia symptoms, which determined a dynamic change in clinical diagnosis and management, ultimately taking into consideration arrhythmogenic cardiomyopathy. Through the article, we try to uncover and explain common pathophysiological pathways shared by arrhythmogenic cardiomyopathy and other clinical entities with a special focus on inflammation. The final question remains: *“If there is more than one heart disorder that eventually leads to the same clinical image, one may wonder, is arrhythmogenic cardiomyopathy a syndrome rather than a specific condition?”.*

**Figure 1 diagnostics-14-00248-f001:**
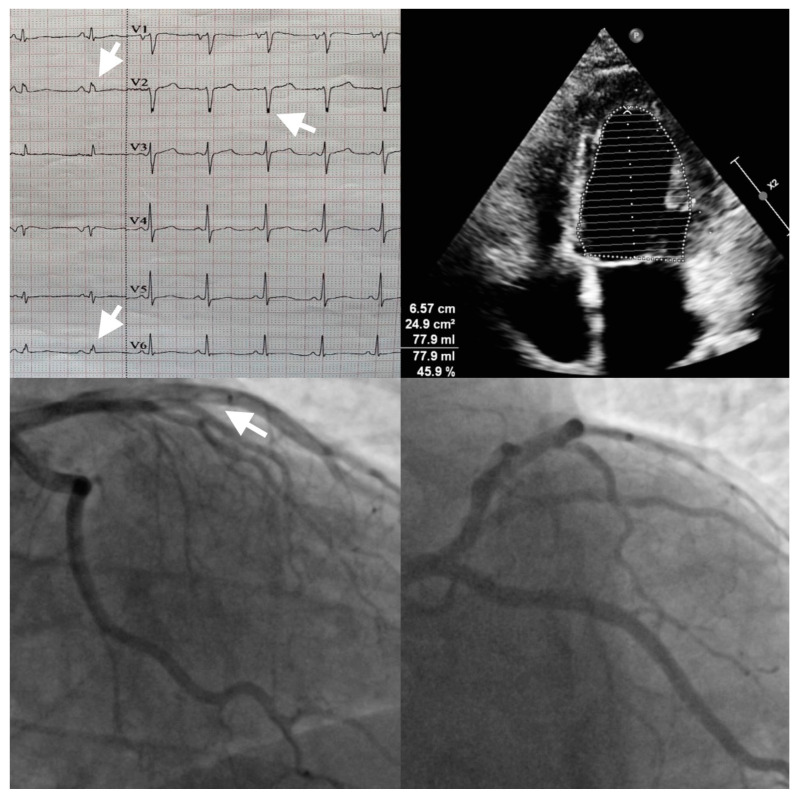
ECG (echocardiography) and coronary angiography: notched R and S waves (arrow on echocardiograph), “slow-flow” phenomenon in left anterior descendent artery (arrow on angiography). A 31-year-old male smoker patient, presented to the emergency department for anterior chest pain with sudden onset and neck radiation. From his personal and family medical history, we did not uncover any relevant cardiovascular health issues. His heart rate, blood pressure and vitals at presentation were in the normal range. The electrocardiogram showed a 2 mm ST-segment elevation in V2–V3 and QRS fragmentation with notching of the R wave in D2 and aVF and of the S wave in V1–V2. The lab exam showed a markedly increased high-sensitive troponin (hsTnI = 16.879 ng/mL) level and normal LDL cholesterol level. Echocardiographic imaging revealed mild systolic dysfunction (EF = 45%) with hypokinesia of the mid-apical septum. The patient was referred immediately to the catheterization laboratory, where coronary angiography showed normal coronary arteries but with a “slow-flow” phenomenon (CSF) in the left anterior descendent artery. Because the patient did not present multiple cardiovascular risk factors and due to the aspect of the “slow-flow” feature seen in the invasive coronary angiography, a diagnosis of myocardial infarction with non-obstructive coronary arteries (MINOCA) with a possible thrombophilia or autoimmune disorder etiology was taken into consideration. The panel tests for autoimmune disorders were normal; instead, the hereditary profile for thrombophilia revealed a positive mutation for factor V Leiden and a heterozygote genotype mutation of the MTHFR gene at the level of the C677T and A1298C loci. In the MTHFR gene, the C677T (rs 1801133) mutation occurs in exon 4, with the substitution of C into T at position 677 and is due to the transformation at the codon 222 *N*-terminal domain of alanine to valine, while A1298C (rs 1801131) mutation occurs in exon 7 due to the change from glutamic acid into alanine at the codon 429 C-terminal domain. The combination of both mutations is associated with an increased level of homocysteine and a low level of folate. Hyperhomocysteinemia is a recognized risk factor for arterial and venous thrombosis. Carriers of the A1298C allele have been reported to have an increased risk of coronary disease. Other associated risk factors (immobilization, trauma, pregnancy, oral contraceptives, smoking, obesity, inflammation) can trigger thrombosis [1,2]. The patient was discharged with the following pharmacological treatment: aspirin, apixaban, non-dihydropyridine calcium channel blockers and long-acting nitrates.

**Figure 2 diagnostics-14-00248-f002:**
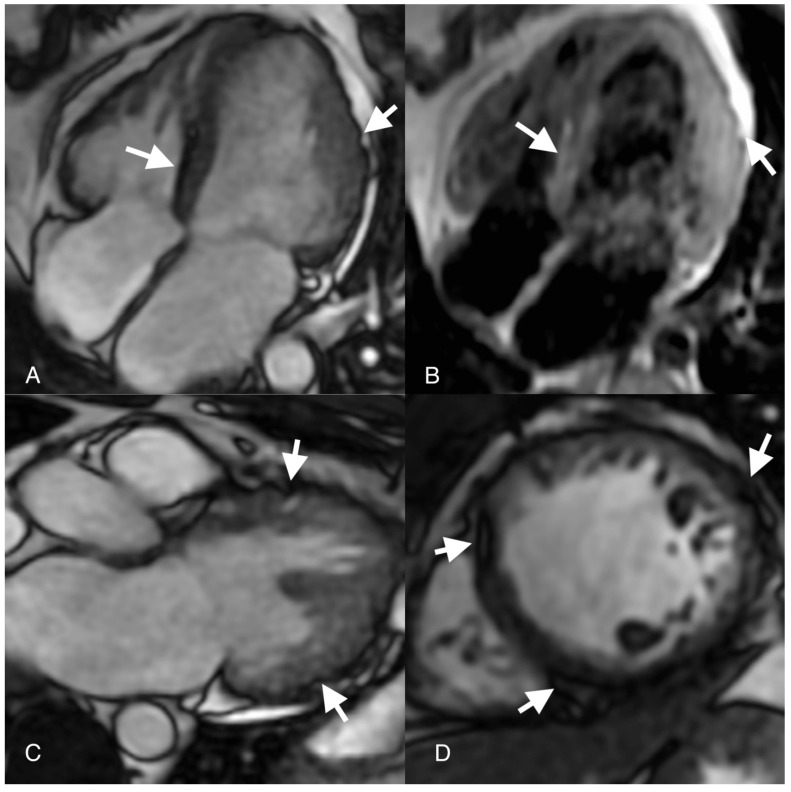
Cardiovascular magnetic resonance (CMR): steady-state free precession (SSFP) images ((**A**,**C**,**D**) arrows) show “chemical shift” and micro-aneurisms; T1-weighted image (**B**) shows (arrow) fat infiltration. After two weeks, the patient presented in the ambulatory office with exertional dyspnea and high levels of natriuretic peptides (NT-proBNP = 924 pg/mL). A cardiac MRI was performed, which showed a dilated left ventricle with moderate systolic dysfunction (EF = 35%), mid-apical anterior wall hypokinesia and septal, inferior and lateral wall hypokinesia of the apical segments. In the CINE images, a “chemical shift” was depicted at the basal segment of the septum and of the apical segment, suggestive of fat metaplasia. Fat infiltration was also observed in T1-weighted and T2-weighted sequences at the level of the basal segment of the septum and lateral wall. The presence of micro-aneurysms was spotted at the level of the mid-apical segments of the lateral wall. According to the “Padua Criteria” for diagnosing arrhythmogenic cardiomyopathy (ACM), morpho-functional and structural MRI lesions were suggestive of dominant left ventricular arrhythmogenic cardiomyopathy [3]. The etiology of ACM is heterogenous as a genetic disorder caused by homozygous mutations of intercellular adhesion molecules, namely the desmosomes. There are several other mutations linked to ACM, such as nuclear envelope, intermediate filament, sarcomere or SERCA mutations, but desmosome abnormalities are by far the most incriminated in this myocardial disease. To our knowledge, this is the first clinical case to highlight ACM features in a patient with an initial diagnosis of MINOCA.

**Figure 3 diagnostics-14-00248-f003:**
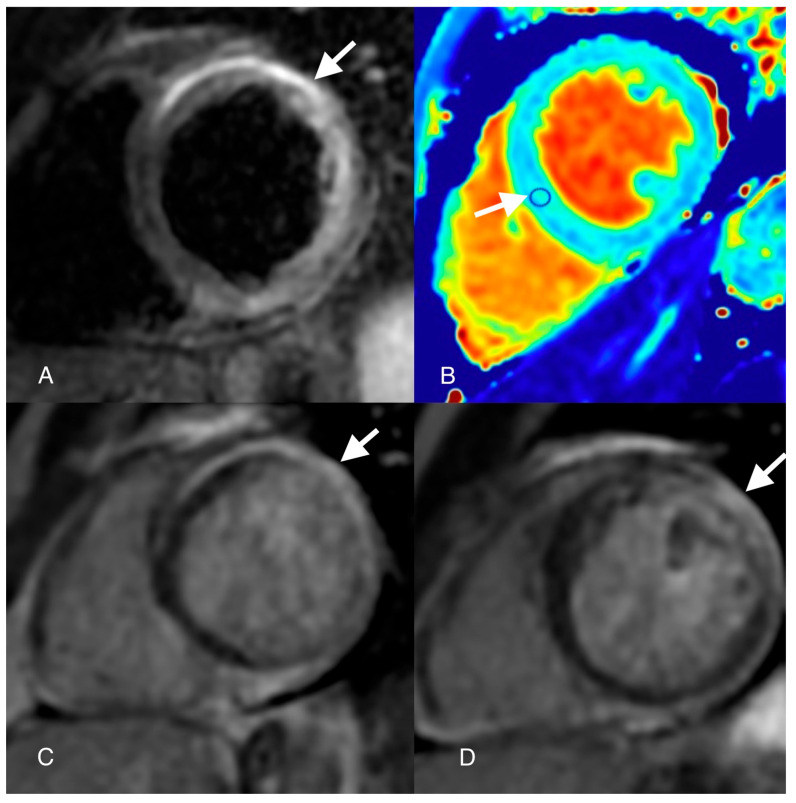
Cardiovascular magnetic resonance (CMR): T2-weighted Triple Inversion Recovery image ((**A**) arrow) shows epicardial anterior and lateral wall high-signal-intensity areas; native T1 map image ((**B**) arrow) shows high T1 values (1400 ms) in septal region of interest (ROI); late gadolinium enhancement (LGE) images ((**C**,**D**) arrows) show epicardial late enhancement of the antero-septal, antero-lateral and infero-lateral wall basal and mid segments. Epicardial myocardial oedema was seen at the basal segment of the anteroseptal and inferolateral walls in the T2-weighted images with a regional SI ratio of the myocardium over the skeletal muscle of 2. Native T1 mapping showed high T1 values (1400 ms) suggestive of myocardial edema. In LGE images, epicardial fibrosis was described in the anterior wall mid-apical segments, inferior wall mid-basal segments, septum and lateral wall segments. Besides the “Padua Criteria”, the patient’s MRI lesions according to the “Lake Louise” criteria were also suggestive of acute myocarditis. The pathophysiology of ACM is rather complex and involves multiple signaling pathway perturbations like plakoglobin redistribution, gap junction remodeling, myocardial apoptosis and high-circulating levels of proinflammatory cytokines [4]. Inflammation secondary to viral infection has been suggested to influence the ACM pathophysiology, as initial histological reports showed patchy inflammatory infiltrates in patients with ARVC [5]. While this may be, there were also a lot of reports of misdiagnosed ACM considered to be myocarditis [6]. The highly phenotypical resemblance between these conditions opened a “Pandora’s box” of discussion whether inflammation triggers ACM or it is just a secondary mechanism of a maladaptive immunomodulatory response involving cytokine-encoding genes of ACM. Although local interstitial right ventricle (RV) and left ventricle (LV) inflammation infiltrates were demonstrated in several studies on ACM cases, this pattern is not universally found in ACM histology [4]. The presence of inflammation in ACM hearts varied from 30% to 67%, involving most often lymphocytes, monocytes, neutrophils and less often mast cells, macrophages and histiocytes in areas of fibro-fatty replacement [7,8]. On the other hand, fibro-fatty replacement is not a characteristic feature of myocarditis pathology. There are frequent reports in the literature where ACM patients clinically present with an acute phase of “myocarditis-like” features (dyspnea, chest pain, elevated troponin levels, ECG abnormalities) [9]. Martins et al. showed in a pediatric population with different variants of ACM the presence of an “acute phase” in six patients of the entire cohort, neither of them being identified with an infectious trigger but rather it being exercise-induced [10]. There are also some small studies that demonstrated even the presence of cardiotropic viruses in biopsies of ARVC hearts [11]. Inflammation was imagistically evaluated in two small sample studies of patients with ACM using myocardial scintigraphy and 18-FDG PET scanning [12,13]. There was a significantly higher myocardial uptake of radiotracers in the ACM group compared to the control group, thus indicating the presence of inflammation [12,13]. Two cases with ACM and confirmed desmoplakin (DSP) mutation were described as presenting with an acute phase episode of a myocarditis-like phenotype and subepicardial LGE using CMR after sports activity [14]. Interestingly, it only involved the LV, and no other imagistic criteria for ACM were met [14]. In human heart biopsies, there was a trend of inflammatory infiltrates associated with fibro-fatty replacement in the ACM LV-dominant form (75%) compared to the RV-dominant form (30%) [7]. Also, these patients showed similar patterns to patients with myocarditis, meaning they had sub-epicardial scarring on the inferolateral wall. But inflammation is not always linked to an infectious trigger, as ACM patients were described as experiencing “myocarditis-like” episodes with inflammation depicted using CMR in the absence of a viral disease [15]. There is evidence that inflammation is described more often in desmosomal forms of ACM rather than non-desmosomal forms. Quite interestingly, when it comes to inflammation and autoimmunity, there are reports that the overexpression of desmocollin-2 and mutations in the desmoglein-2 genes in murine models modulate the inflammatory and repair processes, thus signifying a final common pathway for all ACM subjects. One possible explanation resides in the fact that either genetically or non-genetically determined desmosomal disruption might trigger an immune chain reaction, leading to cell death and amplified anti-desmosomal antibodies [16,17]. These episodes are now recognized as the “hot phases” of ACM and are considered to be responsible for disease progression [9].

**Figure 4 diagnostics-14-00248-f004:**
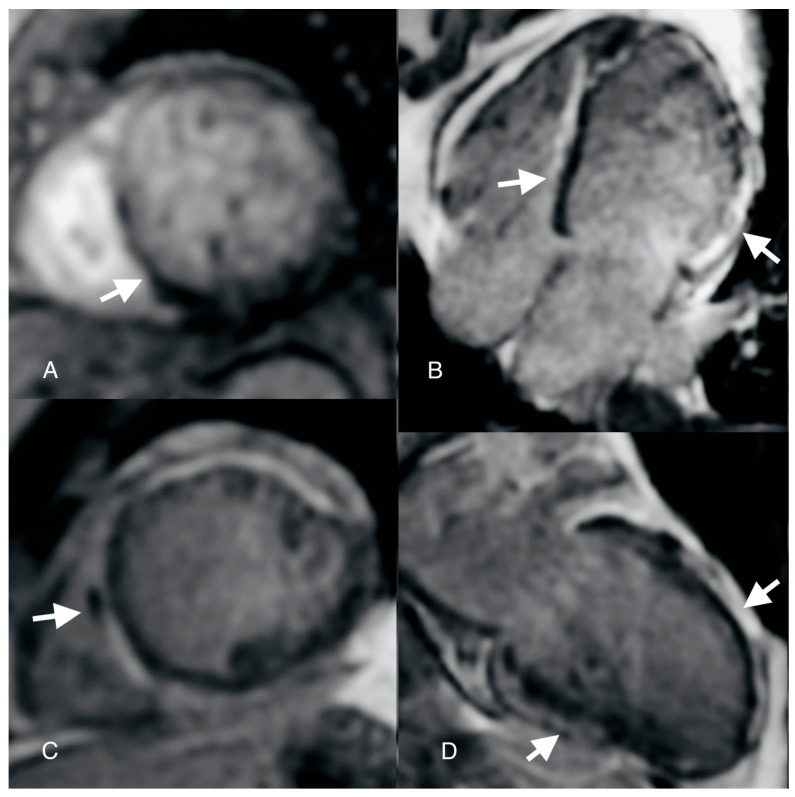
Cardiovascular magnetic resonance (CMR): first-pass perfusion image ((**A**) arrow) shows an inferoseptal perfusion defect; late gadolinium enhancement image ((**C**) arrow) shows a low-signal-intensity area at the level of the septum mid segment; horizontal (**B**) and vertical (**D**) long-axis late gadolinium enhancement (LGE) images show epicardial late enhancement of the anterior, lateral, inferior and septum walls. Also, at the level of the septum mid segment, a hypointense signal was depicted in the LGE sequences suggestive of microvascular obstruction (MVO). A first-pass perfusion was performed which evidenced an inferoseptal perfusion defect in concordance with the “slow-flow phenomenon” described at coronary angiography in the emergency setting. There is, however, a particular aspect regarding this case, mainly the “slow-flow” phenomenon seen using coronary angiography, which created some confusion regarding the patient’s management. This term is mainly used in invasive coronary angiography to describe the delayed progression of contrast media through the coronary artery tree during injection. This phenomenon is frequently encountered in acute coronary syndromes and is due to microvascular obstruction. While the patient was positive for thrombophilia, in the context of CSF, coronary thromboembolism was a reasonable mechanism thought to be responsible for an acute presentation. But there are many underlying pathogenic mechanisms responsible for this phenomenon besides obstruction, such as small vessel disease, endothelial dysfunction or inflammation [18]. It is known that myocarditis may be accompanied by impaired coronary microcirculation. Active pro-inflammation, together with a damaged endothelium, causes a predisposition to thrombus formation and aberrant microvascular vasomotor function, which might explain the CSF phenomenon [19]. But infectious agents are not solely responsible for changes in the microvasculature, such changes being also observed in non-viral mechanisms. In support to this comes a small study by Paul et al. on 10 patients with arrhythmogenic right ventricular cardiomyopathy (ARVC) in which it was shown using PET scans that there is a reduced hyperemic myocardial blood flow and increased coronary vascular resistance [20]. Inflammation and microvascular dysfunction are common mechanisms in true ischemic MINOCA, myocarditis and ACM, which is why workflow diagnosis in MINOCA should include specific testing for the last two entities, especially in patients with arrhythmic burden, to avoid possible catastrophic events. The patient was initiated with heart failure treatment: neprilysin inhibitors, beta blockers, dapaglifozin and spironolactone. After one month, the patient presented an episode of syncope while at home reading. A 24 h Holter ECG was placed, which recorded two episodes of non-sustained ventricular tachycardia. The patient underwent intracardiac defibrillator implantation. After 3 months of optimal heart failure treatment, the ejection fraction remained 35% with no obvious improvement while the patient remained in NYHA class II. The clinical overlap between these two entities, myocarditis and ACM, is quite clear in certain circumstances, making things in the acute phase indistinguishable. That is why, according to the new proposed Padua Criteria, there is need for genetic testing to differentiate between these two entities [3]. The main limitations of this case were a lack of performing genetic testing and endomyocardial biopsy with PCR analysis for viral agents. The fact that the patient presented three consecutive and different clinical pictures, a “chest-pain”-like episode followed by a heart failure-like episode and finally an arrhythmia event, led us to believe that we were dealing with an acute episode of myocarditis, evolving toward auto-reactive myocarditis and inflammatory cardiomyopathy. Of course, in this situation, things remain unclear on whether this was a case of acute myocarditis or an ACM ALVC variant with a “hot phase”. One should bear in mind that with genetic testing, according to the available literature, a specific causative desmosomal gene mutation has only been found in 50% of cases [21]. One thing is for certain: inflammation present in both entities, ACM and myocarditis, and there are common pathways which produce an overlap and misdiagnosis. Future studies are needed to understand the behavior of the inflammatory pathways in these diseases because targeting inflammation might prevent disease progression and impact the outcomes. It is quite clear the clinical overlap between these two entities, myocarditis and ACM, in certain circumstances make things in the acute phase indistinguishable. That is why there is need for genetic testing to differentiate between these two entities. Another important key message is that all patients with MINOCA should undergo cardiac MRI to exclude acute myocarditis. Whether inflammation acts as a primary trigger or a secondary cause of the event it is not well established, but one cannot help but wonder: if two things look the same and act the same, what do you call them?

## Data Availability

Data are unavailable due to privacy and ethical restrictions. The manuscript was written in accordance with the Helsinki Declaration and approved by the hospital’s local ethical committee.

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
