# Peer review of "Inflammation—A Possible Link between Myocarditis and Arrhythmogenic Cardiomyopathy"

_diagnostics, 2024, doi:10.3390/diagnostics14030248_

Round 1

Reviewer 1 Report

Comments and Suggestions for Authors

It is a very well written manuscript. The quality of figures is very high.

But the structure needs to be improved. All the figures are merged with the main text and this make very difficult the reading of the manuscript.

Author Response

Please see attachement.

Reviewer 2 Report

Comments and Suggestions for Authors

In the manuscript 'Inflammation a Possible Link between Myocarditis and Arrhythmogenic Cardiomyopathy' submitted by Lala and Pop-Moldovan to Diagnostics, the authors describe interesting cases of patients with ACM.  However, the manuscript needs a strong revision.

1.) COuld you include histology staining immune cells in explanted myocardial tissue?

2.) It is known that even genetic mouse models (transgenic democollin-2, DSC2) develop ACM in combination with inflammation. I would discuss this point including relevant citations.

3.) Please write all human gene names in Italics.

4.) Please explain the impact of the MTHFR gene in detail?

5.) Line 42: Please indicate the nucleotide changes of these mutations. Please explain their classification according the the ACMG guidelines (Richards et al. 2015).

6.) Please explain in a material and method section how genotyping was done in detail?

7.) I would explain within the introduction shortly the genetics and pathophysiology of arrhythmogenic cardiomyopathy. Relevant review articles might be relevant in this context.

8.) A Institutional review board statement should be indicated!! Please indicate also that you followed the Declaration of Helsinki.

In summary, this manuscript has to be improved and expanded at several points before it should be accepted for publication. Therefore, I suggest a major revision.

Reviewer 3 Report

Comments and Suggestions for Authors

Lala et al. present a case report of a young male patient presented as MINOCA that was later considered an arrhytmogenic cardiomyopathy.

The authors describe that there were "arrhythmia symptoms". There is no description of rhythm disorders in the text. As arrhythmia is one of the main components of the described cardiomyopathy I believe it is crucial to include a detailed description.  Was Holter ECG monitoring performed? 

The authors say that there was "a dynamic change in patient management". However, there is no description of the treatment of the patient later in the text. How did the MRI findings change the patients' management? How would you manage a patient with AVC and another with myocarditis?

What are the most important key points to learn from this case?

Author Response

Please see attachement.

Round 2

Reviewer 2 Report

Comments and Suggestions for Authors

The authors have improved their manuscript according to the suggested points. In my view the manuscript should be accepted for publication. 

Reviewer 3 Report

Comments and Suggestions for Authors

The authors have addressed my comments and the manuscript has improved substantially.